# Severe Asthma or Chronic Obstructive Pulmonary Disease with Eosinophilic Inflammation? From Uncertainty to Remission under Anti IL-5R Therapy

**DOI:** 10.3390/medicina60030387

**Published:** 2024-02-25

**Authors:** Bianca Oprescu, Oana Raduna, Stefan Mihaicuta, Stefan Frent

**Affiliations:** 1Pulmonology Department, Infectious Diseases and Pulmonology Hospital “Victor Babes” Timisoara, 300310 Timisoara, Romania; biancastepan3@gmail.com (B.O.); stefan.mihaicuta@umft.ro (S.M.); frentz.stefan@umft.ro (S.F.); 2Center for Research and Innovation in Precision Medicine of Respiratory Diseases, Department of Pulmonology, “Victor Babes” University of Medicine and Pharmacy Timisoara, 300041 Timisoara, Romania

**Keywords:** severe eosinophilic asthma, benralizumab, phenotype, endotypes, COPD, biological therapy

## Abstract

*Background and Objectives*: Severe adult-onset eosinophilic asthma and COPD with eosinophilic inflammation are two entities with a similar clinical course and are sometimes difficult to differentiate in clinical practice, especially in patients with a history of smoking. Anti-IL-5 or -IL-5R biological therapy has been shown to be highly effective in severe eosinophilic asthma but has not demonstrated significant benefit in patients with COPD with the eosinophilic phenotype. Our aim was to illustrate this issue in the form of a case report. *Materials and Methods*: We present the case of a 67-year-old patient who is a former smoker with late-onset severe uncontrolled asthma (ACT score < 15) who experienced frequent exacerbations requiring treatment with systemic corticosteroids. The patient’s lung function gradually worsened to a nadir FEV1 = 18%, despite a high dose of ICS in combination with a LABA and intermittent courses of OCS, with negative allergic skin-tests, but with high blood eosinophils level. Biological treatment with an anti-IL5R monoclonal antibody (benralizumab) was initiated, despite the difficulty in the differential diagnosis between asthma and COPD with eosinophilic inflammation. *Results*: The patient’s evolution was favorable; clinical remission was effectively achieved with significant improvement in lung function (FEV1 > 100%), but with persistence of residual mild fixed airway obstructive dysfunction (FEV1/FVC < 0.7). The therapeutic response has been maintained to date. *Conclusions*: Benralizumab was shown to be very effective in a patient with late-onset severe eosinophilic asthma presenting features of chronic obstructive disease—habitual exposure to tobacco and inhaled noxious substances, and persistent airflow limitation on spirometry.

## 1. Introduction

Bronchial asthma is a chronic obstructive airway disease characterized by inflammation and structural changes that most commonly cause reversible airflow limitation and is responsible for recurrent episodes of coughing, dyspnea, wheezing, and chest tightness [1]. Most patients with asthma have a mild or moderate disease. There is large geographical variation in asthma epidemiology reports, with higher prevalence in high-income countries and most asthma-related mortality occurring in low-income countries [2,3]. Data on the prevalence of severe asthma are sparse, but reports indicate that between 5 and 10% of the asthmatic population has severe disease [4].

Severe asthma is defined as asthma that remains uncontrolled despite optimized treatment with high-dose inhaled corticosteroids and long-acting bronchodilators (ICS/LABA), or requires such treatment to prevent it from becoming uncontrolled [5]. It is associated with high risk of severe exacerbations and death, and is responsible for a high proportion of the total costs for asthma care [4]. Uncontrolled asthma is not always synonymous with severe disease. Poor control may depend on refractoriness to treatment, but may also be caused by other factors such as poor treatment adherence, which may be due to costs, side effects, and an inability to understand how to use inhalers or negligence. Indicators of poor asthma control are high consumption of rescue medication (SABA), the need for repeated oral corticosteroid (OCS) bursts, visits to the emergency room, and hospitalizations because of asthma [6].

This disease affects more than 300 million people worldwide and is based on complex interactions between genetic and environmental factors [7]. To better understand this heterogeneity, the concept of asthma phenotyping has emerged. Phenotyping integrates biological and clinical features, ranging from molecular to cellular, morphological, functional, and patient-oriented characteristics, with the goal to improve therapy [6].

Asthma phenotypes are classified according to the type of airway inflammation through the assessment of induced sputum as eosinophilic (Eos > 2–3%), neutrophilic (Neu > 61–76%), mixed (both Eos and Neu elevated) and paucigranulocytic (Eos < 2–3% and Neu < 61%), and also according to the clustering of clinical, functional, and inflammatory profiles as follows: early-onset mild allergic asthma (Th2 eosinophilic cluster), early-onset allergic moderate-to-severe remodeled asthma (Th2 mixed granulocytic cluster), late-onset nonallergic eosinophilic asthma (T2 eosinophilic cluster), and late-onset non-eosinophilic non-allergic asthma (non-T2 neutrophilic cluster) [6].

Ultimately, these phenotypes should evolve into asthma “endotypes”, known as T2-high and T2-low asthma, which combine clinical characteristics with identifiable mechanistic pathways [7]. The T2-high phenotypes share an immune–inflammatory response driven by Th2 lymphocytes (adaptive immunity) and group 2 ILCs (ILC2, innate immunity). The hallmarks of T2 inflammation are T2 cytokines such as IL-5, IL-4, IL-13, IL-9, prostaglandin D2 (PGD2), and eosinophils, whose high expression can be detected in the airways (bronchial lumen or wall) and peripheral blood of patients. Eosinophilic asthma is considered a synonym of T2-high asthma, encompassing both allergic and nonallergic phenotypes; it is described in 50% of patients with severe asthma and is characterized by the presence of blood eosinophilia (>300 cells/mmc). T2-low asthma encompasses all forms of asthma lacking the distinctive traits of T2-high asthma: neutrophilic and fibrosis-associated, and linked with the expression of IL-17, IL-8, and fibroblasts [7,8]. The most important cytokine involved in the induction, maintenance, and amplification of eosinophilia in asthma is interleukin-5 (IL-5), released by both T helper 2 (Th2) lymphocytes and innate lymphoid group 2 cells (ILC2). Biological therapies that target this interleukin or its receptor are now available: mepolizumab and reslizumab—anti IL-5 monoclonal antibodies—and benralizumab—an anti IL-5R monoclonal antibody. The latter is an antibody directed against the α subunit of the interleukin (IL)-5 receptor (IL-5Rα), which depletes eosinophils via antibody-dependent cell-mediated cytotoxicity (ADCC) [9,10]. Other biological therapies approved for severe asthma treatment are omalizumab, a monoclonal anti-IgE antibody that binds specifically to circulating IgE molecules, thus interrupting the allergic cascade, that has been shown to be highly effective in treating children and adults with moderate-to-severe allergic asthma [11]; dupilumab, a monoclonal antibody that binds to the IL-4α receptor, blocking IL-4 and IL-13 intracellular signaling, which results in the reduced expression of proinflammatory cytokines, ultimately leading to decreased total and specific IgE, decreased FeNO, and a transient increase in blood eosinophils [12]; and more recently, tezepelumab, which is a TSLP alarmin-blocking drug that inhibits the initiation of the T2 inflammatory cascade from both the innate (via ILC2) and adaptive (via Th2) immune response, and also possibly the neutrophilic non-T2 pathway (via IL-17), showing promising results in clinical trials [13].

On the other hand, an eosinophilic phenotype is also described in COPD, which has several unique features, such as a specific pattern of bronchial inflammation, a distinct clinical course, and susceptibility to corticosteroid treatment [14]. Blocking IL-5 using different biological agents has been shown to be effective in attenuating T2-type inflammation in severe eosinophilic asthma, but not in COPD patients with eosinophilic inflammation [15,16,17].

## 2. Case Description

We present the case of a 67-year-old patient who is an ex-smoker (10 pack year) with occupational exposure to inhaled noxious substances (exhaust fumes, about 30 years) and biomass fuel, with no evidence of other chronic diseases or allergies. The patient had been diagnosed in the previous year with moderate asthma, based on evidence of fixed moderate obstructive ventilatory dysfunction and a postbronchodilator increase in FEV1 > 200 mL and >12%, and had been prescribed inhaled treatment with a fixed dose of an ICS/LABA combination—HFA beclomethasone/formoterol, 100/6 mcg, one puff b.i.d. The patient presented in May 2019 to the Pulmonology Department with symptoms of exacerbation: persistent, productive cough with mucous sputum, dyspnea on minimal effort (mMRC = 2), intermittent wheezing, nocturnal awakenings (ACT score = 14), and serous rhinorrhea; lung auscultation detected wheezing and bilaterally disseminated bronchial rales. Spirometry was performed, showing severe fixed obstructive ventilatory dysfunction (FEV1% = 42%, FEV1/FVC = 0.46) with significant reversibility post bronchodilation (140 mL and 12%). We decided to step up the inhaled treatment to a medium dose of ICS/LABA—beclomethasone/formoterol, 100/6 mcg, two puffs b.i.d., and recommended a course of oral corticosteroids (prednisone for 9 days, with tapering doses).

He was referred for an allergy consultation, and the skin prick test yielded negative results to common aeroallergens. After one month (June 2019), the patient was seen in the Ambulatory Clinic for control and reported improved respiratory symptoms (ACT score = 19), no rales on lung auscultation, and significant improvement in lung function tests. Spirometry showed fixed mild obstructive ventilatory dysfunction with reversibility of 290 mL and 12% post-bronchodilator (BD) treatment (FEV1% = 87%, FEV1/FVC = 0.51). 

Under inhaled treatment with medium-dose ICS/LABA, an initial favorable clinical and functional response was observed, followed by progressive worsening of the disease, despite adherence to treatment and an appropriate inhalation technique (Figure 1 and Figure 2). 

The following period was marked by successive episodes of exacerbations requiring hospitalization or repeated courses of systemic corticosteroids.

In February 2020, he presented with worsened respiratory symptoms in the context of a most likely viral respiratory infection: productive cough with mucous sputum, dyspnea, wheezing, nocturnal awakenings (ACT score = 12), and the frequent use of rescue medication (SABA); clinical examination revealed wheezing and bilaterally disseminated bronchial rales, and spirometry showed severe fixed obstructive ventilatory dysfunction without post-bronchodilation reversibility (FEV1% = 36%, FEV1/FVC = 0.36). With the evidence for the lack of asthma control, we decided to again step up the inhaled treatment to high dose of beclomethasone/formoterol, 100/6 mcg, two puffs t.i.d., and the patient was prescribed another course of oral prednisone for 9 days, with tapering doses.

Despite these therapeutic measures, in April of 2020, the patient presented with an episode of severe exacerbation with acute respiratory failure, requiring hospitalization. He tested negative for SARS-CoV-2 infection; the chest X-ray showed a bilateral marked interstitial pattern; an echocardiogram was performed and several cardiovascular comorbidities were diagnosed: degenerative mitral and aortic regurgitation and severe functional tricuspid regurgitation in the context of severe secondary pulmonary hypertension. He was treated with oral antibiotics, systemic corticosteroids, mucolytics, bronchodilators, and ICS, with an improvement in his respiratory condition.

After only 3 months, in July 2020, the patient presented to our department complaining of worsened asthma symptoms: severe dyspnea at rest (mMRC = 3) with tachypnea, unproductive cough, wheezing, and nocturnal awakenings (ACT score = 7); his symptoms improved following oral prednisone treatment and returned on discontinuation. The patient had been prescribed inhaled add-on tiotropium, but self-discontinued upon intolerance. Clinical examination revealed a hyperinflated chest, wheezing, bilaterally disseminated bronchial rales, SpO_2_ = 90% spontaneously, and elevated blood pressure values (180/120 mmHg). The laboratory parameters showed leukocytosis with neutrophilia, marked eosinophilia (1030 cells/mmc), while the spirometry revealed very severe fixed obstructive ventilatory dysfunction (FEV1% = 18%, FEV1/FVC = 0.27). As the patient’s condition did not improve despite the high dose of inhaled beclomethasone/formoterol, we decided to add 10 mg of prednisone/day as maintenance treatment on top of the inhaled medication.

Although the patient was on treatment with OCS to control the clinical symptoms and reduce hypereosinophilia, and despite an adequate inhalation technique and good adherence to high doses of the inhaled corticosteroid and bronchodilator treatment, the trend of clinical, biological, and functional deterioration was maintained. At this point, the necessity to distinguish the predominant phenotype of the airway disease in this patient became very important. On one hand, the patient had features of COPD, like an onset age of disease of higher than 40 years and exposure to inhaled noxious particles, tobacco smoke, and biomass fuels, with respiratory symptoms worsening over time and persistent airflow limitation (post-BD FEV/FVC < 0.7). On the other hand, the extremely high variability of symptoms and functional tests over time, as well as a good response to ICS/OCS, were the decisive arguments for the diagnosis of asthma. The presence of hypereosinophilia > 1000 cells/mmc required a differential diagnosis of a parasitic infection, in particular, helminthiasis, and eosinophilic granulomatosis with polyangiitis. Due to the rapid succession of exacerbation episodes and worsening of the patient’s clinical status on one hand, and due to the immense pressure exerted by the COVID-19 pandemic on the healthcare system in general, and to our hospital in particular, on the other, the two conditions were excluded based on clinical grounds. Our country is not endemic for helminth infection, and the patient did not travel abroad and did not report any gastro-intestinal symptoms suggestive for helminthiasis. The patient also did not report any symptoms related to the upper respiratory tract, such as rhinitis, nasal polyposis, or sinusitis. The chest X-ray did not demonstrate the presence of pulmonary infiltrates and no physical signs of vasculitis or neuropathy were noticed. Thus, the decision was taken to start biological treatment with an anti-IL-5R monoclonal antibody—benralizumab, 30 mg/month in the first 2 months, and then, 30 mg every 2 months.

In July 2020, we administered the first dose of benralizumab via subcutaneous injection, with no reporting of adverse reactions. The patient continued the inhaled treatment with a high dose of ICS/LABA and 5 mg/day of prednisone. The start of biological therapy marked a turning point in the evolution of the disease. The clinical status of the patient started to improve immediately, no further exacerbations were experienced, and the maintenance therapy was gradually stepped down. Two months after the initiation of benralizumab, the patient assessment revealed an extremely favorable clinical and functional response (ACT score = 22, FEV1% = 86%), and the maintenance OCS was withdrawn; by the third dose of biological treatment, the blood eosinophils were completely depleted (0 cells/mmc) (Figure 3).

Currently, the patient’s favorable evolution is maintained > 3.5 years after the initiation of benralizumab; he reports not having any respiratory symptoms (ACT score = 25), with complete cessation of exacerbation episodes, depleted levels of blood eosinophils, and spirometry showing mild fixed obstructive ventilatory dysfunction (FEV1% = 116%, FEV1/FVC = 0.64); he continues inhaled treatment with a medium dose of beclomethasone/formoterol, 100/6 mcg, two puffs b.i.d.

## 3. Discussion

In this case report, we described the efficacy of benralizumab treatment in a patient with severe eosinophilic asthma, in whom a diagnosis of COPD cannot be ruled out. Clinical remission of severe asthma was effectively obtained in this case; however, residual mild fixed obstructive ventilatory dysfunction may be suggestive of the co-existence of COPD. 

The patient presented with a history of frequent exacerbations and progressive worsening of clinical status and lung function, despite maintenance treatment with high-dose inhaled ICS/LABA and intermittent courses of oral corticosteroids (OCS). The introduction of biological therapy with an anti-IL5R drug radically changed the clinical course of the disease in this super-responder patient.

There are several particularities that deserve discussion in this case.

First, it is difficult to make a differential diagnosis between asthma and COPD. Second, there is a surprising evolution toward progressive worsening of the disease despite adequate management and adherence to the inhaled treatment. Third, there is a close connection between the clinical evolution of the respiratory disease and cardiac comorbidities.

Distinguishing between asthma and COPD is not always a straightforward process, especially in patients with common features for both conditions, such as onset of the airway disease at adult age (>40 years), a history of smoking and/or exposure to inhaled noxious particles, or evidence of fixed airway obstruction combined with a reversibility of FEV1 of ≥ 200 mL and ≥12%. In our patient, both asthma and COPD can be strongly supported and, in our opinion, the two conditions co-exist, although the predominant phenotype is asthma. The difficulty in establishing the predominant phenotype reached its highest when the patient’s condition was at its worst, with continued, progressive deterioration due to successive exacerbation episodes and collapsed lung function (FEV1% = 18%). Previous studies demonstrated an excellent response to biological therapy with anti-IL5R agents in patients with severe eosinophilic asthma [18,19], but not in COPD patients with eosinophilic inflammation [17]. The strongest argument in favor of asthma and for the decision to start the biological treatment with benralizumab was the extremely high variability in lung function parameters (i.e., FEV1), both during worsening as well as during the remission period. The spectacular clinical and functional response confirmed we were right in our reasoning.

Currently, there are three available biological treatments for severe asthma with hypereosinophilia—mepolizumab and reslizumab—two monoclonal antibodies against IL-5—and benralizumab, a monoclonal antibody against the IL-5 receptor, which is the only one available in our country. Benralizumab has been shown to be effective in controlling both clinical symptomatology (ACT score = 25) and exacerbations, and in completely depleting circulating eosinophils within the first 2–4 months of treatment. In terms of functional measures, a remarkable increase in FEV1 was observed, from 18% before treatment initiation to 120% after 2 years of treatment, but with persistence of fixed obstructive ventilatory dysfunction on serial spirometry, with FEV1/FVC values constantly < 0.7. This is a strong argument for the co-existence of COPD, especially when taking into account the patient’s age, smoking history, and occupational exposure to inhaled noxious substances. Despite the fact that benralizumab has not been shown to be effective in COPD with eosinophilic inflammation, in this case, sustained remission 3 years after treatment initiation was effectively achieved, which is a strong argument for the predominance of an asthma phenotype.

To conclude, severe adult-onset eosinophilic asthma and COPD with eosinophilic inflammation are two entities with a similar clinical course, and are sometimes difficult to differentiate in clinical practice, especially in patients with a history of smoking, and perhaps their co-existence is likely possible in the same patient.

Another interesting particularity in this case is that the clinical course of the disease surprisingly underwent significant worsening despite adequate therapeutic management, with successive exacerbation episodes, mirrored by a significant increase in the number of blood eosinophils (>1000 cells/µL). It is not at all discounted that the outcome would have been fatal if we had not introduced biological therapy with anti-IL5R. This feature is more typical for COPD, whereas asthma tends to have a more stable pattern of severity.

Finally, it also worth mentioning that some of the cardiovascular comorbidities diagnosed during the episode of exacerbation requiring hospitalization, including secondary pulmonary hypertension, improved significantly with the remission of severe asthma symptoms. This aspect emphasizes the strong and bidirectional connections between the lung and the heart.

Currently there is a lack of data establishing the duration of biological therapy or dose reduction after sustained remission. Given the cost of such therapy, as well as the frequent visits of patients to outpatient clinics, the question arises as to whether we should consider extending the time intervals between administrations, while maintaining the same results over time? On the other hand, the advantage of biological therapies is that they address the various mechanisms underlying the pathophysiology of asthma, and therefore, their discontinuation is likely to be followed by worsening of symptoms, inflammation, bronchial obstruction, and blood eosinophilia.

## 4. Conclusions

Biological therapy in severe asthma, now available in Romania, is highly effective in selected, carefully phenotyped cases, with the correct diagnosis being essential for the success of this type of therapy.

## Figures and Tables

**Figure 1 medicina-60-00387-f001:**
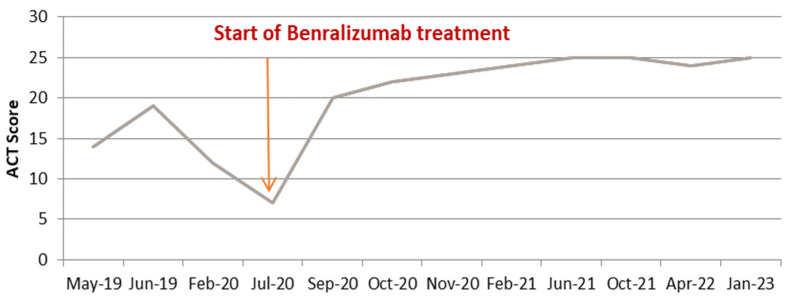
Asthma control as assessed by ACT score before and after anti IL-5R treatment.

**Figure 2 medicina-60-00387-f002:**
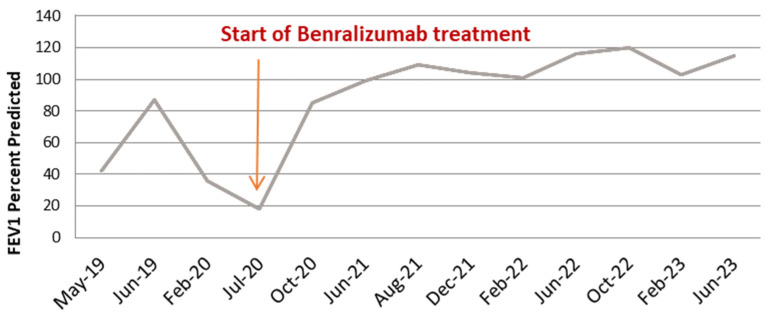
Lung function as assessed by FEV1 before and after anti IL-5R treatment.

**Figure 3 medicina-60-00387-f003:**
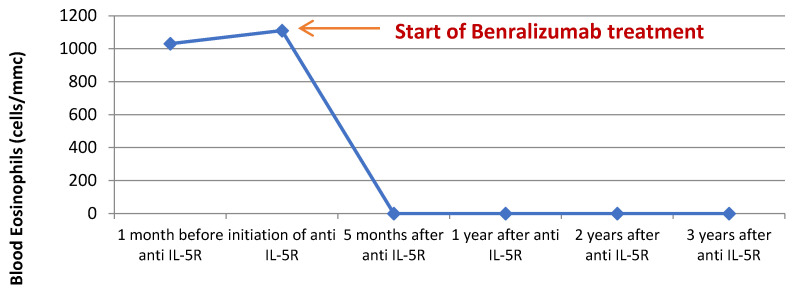
Blood eosinophils before and after initiation of benralizumab treatment.

## Data Availability

The original contributions presented in this study are included in the article. Further inquiries can be directed to the corresponding author.

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
