# Peer review of "Severe Asthma or Chronic Obstructive Pulmonary Disease with Eosinophilic Inflammation? From Uncertainty to Remission under Anti IL-5R Therapy"

_medicina, 2024, doi:10.3390/medicina60030387_

Round 1

Reviewer 1 Report

Comments and Suggestions for Authors

First I would like to congratulate the authors on completion of this manuscript.

Anti IL-5 therapy is increasingly is being used in patients with moderate to severe asthma not responding to standard medium to high dose ICS and bronchodilator therapy, or those with frequent exacerbation needing chronic corticosteroids. there has been recently few studies exploring the role in COPD with higher Eos count.

This patient had late onset asthma (which although seems rare but is not uncommon) with risk factors of COPD. given high Eos count, variability in spirometry and clinical symptoms it is safe to assume he had more of an eosinophilic asthma component driving his airways disease which is likely to respond to anti IL-5 therapy, in this case benralizumab. 

in case description section, please add physical examination findings as well before discussing diagnostic/lab data.

also as part of escalating treatment it is important to ensure inhaler compliance and adequate technique. was that done in this case?

Comments on the Quality of English Language

well written, minor English revision/editing needed

Author Response

Thank you for words of appreciation and the adequate suggestions for improvement of our manuscript! We have, of course, checked at every visit the inhalation technique and the adherence to the prescribed inhaled treatment. This was already mentioned in the manuscript as a generic comment on the line 120 and we have now included a similar comment on the lines 157-158. With regards to the physical examination, we have included the findings after the description of presenting symptoms for each clinical assessment and before discussing the lab data (e.g. lines 105-106, line 115, lines 130-131, lines 149-151 etc.)

Reviewer 2 Report

Comments and Suggestions for Authors

Excellent case report of a 67-years-old patient with late-onset of severe un controlled asthma presented studying ACT Score,  lung function assessed by FEV1 %  and blood eosinophil count before and after anti IL-5R treatment. The authors have also nicely presented the results obtained in the form of graphic representation. However, Figure 3  (blood eosinophils counts) surprisingly shows zero count post 2 months of anti IL-5R  treatment and continued till  observation period of 3 years post treatment. 

Graphic representation of such values some time fails to  reflect the exact values, Therefore, the authors are advised to check the results obtained and show these values on x axis of the graph. 

Author Response

Thank you for the words of encouragement! Please kindly note that all the blood eosinophil values measured after initiation of benralizumab treatment were at 0, which means that the graphic representation is accurate. These results are in line with those reported in the pivotal SCIROCCO, CALIMA and BORA studies. However, the reviewer is right that the timely representation of the measurements is not accurate. We have now amended this error. Thank you for spotting this issue!

Reviewer 3 Report

Comments and Suggestions for Authors

The case, although not exceptional, is illustrative and well-described.

However, I missed two important points:

·       With such a high degree of blood eosinophilia, a wider differential diagnosis of eosinophilia is necessary. In particular, from helminth invasion and eosinophilic granulomatosis with polyangiitis.

·       In the case of severe, difficult-to-control bronchial asthma, additional investigations are necessary, in particular a chest CT scan. Also information on the condition of the upper respiratory tract.

Unfortunately, I could not find this information in the manuscript.

Author Response

Thank you for raising these important issues. Due to the rapid succession of exacerbation episodes and worsening of the patient’s clinical status on one hand, and due to the immense pressure exerted by the COVID-19 pandemic to the healthcare system in general, and to our hospital in particular, the two conditions mentioned by the reviewer were excluded based on clinical grounds. Our country is not endemic for helminth infection and the patient did not travel abroad and did not report any gastro-intestinal symptoms suggestive for helminthiasis. The patient did not report any symptoms related to the upper respiratory tract, such as rhinitis, nasal polyposis or sinusitis. The chest X-ray did not demonstrate the presence of pulmonary infiltrates and no physical signs of vasculitis or neuropathy were observed. We added these clarifications to the manuscript.

Round 2

Reviewer 3 Report

Comments and Suggestions for Authors

None